# CARE: Modeling Interacting Dynamics Under Temporal Environmental Variation

**Xiao Luo**[1], **Haixin Wang**[2], **Zijie Huang**[1], **Huiyu Jiang**[3],
**Abhijeet Sadashiv Gangan**[1], **Song Jiang**[1], **Yizhou Sun**[1]
[1]University of California, Los Angeles, [2]Peking University,
[3]University of California, Santa Barbara
{xiaoluo,yzsun}@cs.ucla.edu, wang.hx@stu.pku.edu.cn

## Abstract

Modeling interacting dynamical systems, such as fluid dynamics and intermolecular interactions, is a fundamental research problem for understanding and simulating complex real-world systems. Many of these systems can be naturally represented by dynamic graphs, and graph neural network-based approaches have been proposed and shown promising performance. However, most of these approaches assume the underlying dynamics does not change over time, which is unfortunately untrue. For example, a molecular dynamics can be affected by the environment temperature over the time. In this paper, we take an attempt to provide a probabilistic view for *time-varying* dynamics and propose a model Context-attended Graph ODE (CARE) for modeling time-varying interacting dynamical systems. In our CARE, we explicitly use a context variable to model time-varying environment and construct an encoder to initialize the context variable from historical trajectories. Furthermore, we employ a neural ODE model to depict the dynamic evolution of the context variable inferred from system states. This context variable is incorporated into a coupled ODE to simultaneously drive the evolution of systems. Comprehensive experiments on four datasets demonstrate the effectiveness of our proposed CARE compared with several state-of-the-art approaches.

## 1 Introduction

Modeling interacting dynamical systems is a fundamental machine learning problem [66, 65, 23, 59, 37] with a wide range of applications, including social network analysis [12, 17, 36] and particle-based physical simulations [46, 43, 35]. Geometric graphs [27] are utilized to formalize these interactions between objects. For example, in particle-based physical systems, edges are constructed based on the geographical distance between particles, which represents the transfer of energy.

In the literature, numerous data-driven approaches have been proposed for understanding interacting dynamical systems [2, 25, 49]. Among them, graph neural networks [26, 62, 72, 33, 13] (GNNs) are widely utilized to predict trajectories at the next timestamp due to their strong capacity to capture interactions in graph-structured data. In particular, each object is considered as a graph node, and edges represent interactions between neighboring objects. Given the observations and their corresponding graph structure, these methods forecast states in the next timestamp using the message passing mechanism. This process involves aggregating information from the neighbors of each node to update its representation in an iterative fashion, effectively capturing the dynamics of the system.

Although impressive progress has been achieved on GNN-based approaches, capturing long-term dependency in interacting dynamical systems is still a practical but underexplored problem. Existing next-step predictors [42, 46, 45] can send the predictions back to generate rollout trajectories, which could suffer from serious error accumulation for long-term predictions. More importantly, system

37th Conference on Neural Information Processing Systems (NeurIPS 2023).

environments and relational structures could be changeable [69] (e.g., unsteady flow [11, 16]), which implies the potential temporal distribution variation during the evolution. In particular, in physical systems, there are various potential factors which can influence the trajectories extensively. For example, high temperatures [69] or pressure [68] could speed up molecular movement. Their continuous variation would make understanding interacting dynamic systems more challenging. First, temporal environmental variation would indicate different data distributions over the time [52], which requires the model equipped with superior generalization capability. In contrast, existing methods typically focus on in-distribution trajectories [7, 1, 58, 3, 21, 66], which would perform worse when it comes to out-of-distribution data. Second, recent out-of-distribution generalization methods [50, 39, 57, 64, 44] usually focus on vision and text with discrete shift across different domains. However, our scenarios would face the continuous distribution variation, which is difficult to capture in interacting dynamical systems.

In this paper, we propose a novel method named Context-attended Graph ODE (CARE) to capture interacting system dynamics. The core of our CARE approach is to characterize the temporal environmental variation by introducing the context variable. In particular, we propose a probability model to depict the interaction between the context variable and trajectories. Based on the probabilistic decomposition, we divide each training sequence into two parts for initializing embeddings and predictions. Here, we first construct a temporal graph and then leverage an attention-based encoder to generate node representations and the context representation from spatial and temporal signals simultaneously. More importantly, we introduce coupled ODEs to model the dynamic evolution of node representations and the context variable. On the one hand, we adopt a graph-based ODE system enhanced with context information to drive the evolution. On the other hand, the context information can also be updated using summarized system states and the current context. We also provide a theoretical analysis that indicates, at least locally, the future system trajectory and context information are predictable based on their historical values. Finally, we introduce efficient dynamical graph updating and robust learning strategies to enhance the generalization capability and efficiency of the framework, respectively. Extensive experiments on various dynamical systems demonstrate the superiority of our proposed CARE compared with state-of-the-art approaches.

To summarize, in this paper we make the following contributions: (1) *Problem Formalization*. We formalize the problem of temporal environmental variation in interacting dynamics modeling. (2) *Novel Methodology*. We analyze the problem under a probabilistic framework, and propose a novel approach CARE, which incorporates the continuous context variations and system states into a coupled ODE system. (3) *Extensive Experiments*. Extensive experiments conducted on four datasets validate the superiority of our CARE. The performance gain of our proposed CARE over the best baseline is up to 36.35%.

## 2   Related Work

**Interacting Dynamics Modeling.** Deep learning approaches have been extensively used in recent years to model interacting systems across various fields [9, 53, 29, 20], including molecular dynamics and computational fluid dynamics. Early efforts focus on incorporating convolutional neural networks to learn from interacting regular grids [41]. To address more generalized scenarios, graph neural network (GNN) methods have been developed [42, 46, 45], leveraging message passing mechanisms to extract complex spatial signals. However, these methods often fail to account for environmental fluctuations, which hinders their ability to make reliable long-term predictions. In contrast, our CARE adopts a context-attended ODE architecture to explicitly represent both the observations and the underlying environment, enabling the generation of accurate future trajectories.

**Neural Ordinary Differential Equations (ODEs).** Drawing inspiration from the approximation of ResNet [5], neural ODEs equip neural networks with a continuous-depth architecture by parameterizing the derivatives of hidden states. Several attempts have been made to increase the expressiveness of neural ODEs [65], including adding regularization [8] and designing high-order ODEs [61]. Neural ODEs have also been incorporated into GNNs, producing continuous message passing layers to avoid oversmoothing [60] and increase the interpretability of predictions [70]. In this study, we employ a graph ODE architecture to capture the continuous nature of interacting system dynamics, relieving potential error accumulation caused by discrete prediction models.

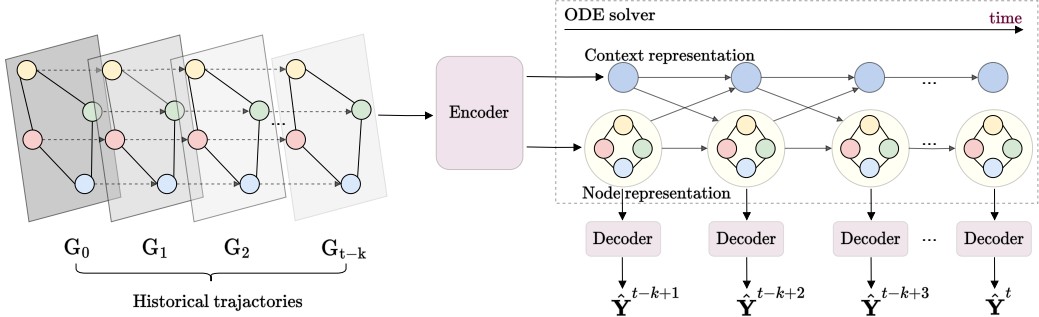

Figure 1: An overview of the proposed CARE. To begin, we construct a temporal graph and utilize an encoder to initialize both the context variable and node representations from historical trajectories. Then, a coupled ODE model simulates the evolution of both nodes and context. Finally, CARE feeds node representations into decoders, which output the predicted trajectories at any timestamp.

**Out-of-distribution (OOD) Generalization.** OOD generation aims to make models more effective when the training and testing distributions diverge [50, 39, 57, 63]. This problem has drawn considerable attention in several areas, including text and vision [48]. One effective solution is to learn domain-invariant representations in the hidden space [31, 55, 30], which has been achieved under the guidance of invariant learning theory [6, 32]. Additionally, uncertainty modeling [34], causal learning [54, 14], and model selection [38, 56] are employed to improve the performance. Interacting systems inherently exhibit dynamic distribution variation caused by environmental changes, an aspect that remains underexplored in the literature. To address this, our paper proposes a novel approach named CARE to model context information from the perspective of a probabilistic model.

## 3 Background

### 3.1 Problem Definition

In a multi-agent dynamical system, the state at time $t$ is represented by $G^t = (V, E^t, \boldsymbol{X}^t)$, where each node in $V$ corresponds to an object, $E^t$ denotes the current edge set, and $\boldsymbol{X}^t$ signifies the node attribute matrix. Specifically, the state vector for each $i \in V$ is given by $\boldsymbol{x}_i^t = [\boldsymbol{p}_i^t, \boldsymbol{q}_i^t, \boldsymbol{a}_i]$, with $\boldsymbol{p}_i^t \in \mathbb{R}^3$ and $\boldsymbol{q}_i^t \in \mathbb{R}^3$ representing the position and velocity, respectively, and $\boldsymbol{a}_i$ representing static attributes. We are given the sequence $\{G^0, G^1, \cdots, G^t\}$ and aim to learn a model that produces the target dynamic states $\boldsymbol{Y}^s (s > t)$ (e.g., velocity), which are part of $\boldsymbol{X}^s$ at the corresponding time. The temporal environmental variation would result in data distribution changes during the evolution of interacting systems. If we utilize $C^{0:t}$ to indicate the dynamical environment factor till timestamp $t$, we have data from variable distribution, i.e., $(G^{0:t}, \boldsymbol{Y}^s) \sim P(G^{0:t}, \boldsymbol{Y}^s | C^{0:t})$. Thus, we must take these changes into account for accurate trajectory predictions.

### 3.2 GNNs for Modeling Dynamical Systems

Graph neural networks (GNNs) are extensively employed in dynamical system modeling to investigate the interactive relationships between objects [42, 46, 45]. These methods typically use the current states to predict the states of nodes at the next timestamp. Specifically, omitting the time notation, GNNs first initialize node representations and edge representations using encoders:

$$\boldsymbol{v}_i^{(0)} = f^v(\boldsymbol{x}_i), \quad \boldsymbol{e}_{ij}^{(0)} = f^e(\boldsymbol{x}_i, \boldsymbol{x}_j), \tag{1}$$

where $f^v(\cdot)$ and $f^e(\cdot)$ are two encoders for node and edge representations, respectively. Then, they utilize two propagation modules to update these representations at the $l$-th layer, i.e., $\boldsymbol{v}_i^{(l)}$ and $\boldsymbol{e}_{ij}^{(l)}$ following the message passing mechanism:

$$\boldsymbol{e}_{ij}^{(l+1)} = \phi^e\left(\boldsymbol{v}_i^{(l)}, \boldsymbol{v}_j^{(l)}, \boldsymbol{e}_{ij}^{(l)}\right), \quad \boldsymbol{v}_i^{(l+1)} = \phi^v\left(\boldsymbol{v}_i^{(l)}, \sum_{j \in \mathcal{N}_i} \boldsymbol{e}_{ij}^{(l+1)}\right), \tag{2}$$

where $\mathcal{N}_i$ collects the neighbours of node $i$. $\phi^e(\cdot)$ and $\phi^v(\cdot)$ are two functions for representation updating. Finally, they generate the target velocity vectors at the next timestamp using a decoder.

# 4 Methodology

In this paper, we propose a novel method called CARE for modeling interacting dynamics under temporal environmental variation. We start by formalizing a probabilistic model to understand the relationships between trajectories and contexts. Based on this foundation, we construct a spatio-temporal encoder to initialize the representations of nodes and contexts. Then, to simultaneously model their evolution, a coupled graph ODE is introduced where node representations are evolved with the guidance of contexts and the states of the context variable are inferred from current trajectories. Additionally, we introduce a regularization term and dynamic graph updating strategies to enhance our framework. An illustration of our CARE can be found in Figure 1.

## 4.1 Probabilistic Model for System Dynamics under Temporal Distribution Drift

In this work, to tackle the challenge brought by temporal distribution drift, we inject a context variable $\boldsymbol{c}^t$ in our dynamic system modeling, which indicates the environment state at timestamp $t$. For example, the context variable could indicate flow speed, density and viscosity in fluid dynamics.

Here, we make two basic assumptions in our probabilistic model.

**Assumption 4.1.** *(Independence-I) The context variable is independent of the sequences before the last observed timestamp, i.e., $P(\boldsymbol{c}^t|\boldsymbol{c}^{t-k}, G^{0:t}) = P(\boldsymbol{c}^t|\boldsymbol{c}^{t-k}, G^{t-k:t})$, where $t-k$ is the last observed timestamp.*

**Assumption 4.2.** *(Independence-II) Given the current states and contexts, the future trajectories are independent of the previous trajectories and contexts, i.e., $P(\boldsymbol{Y}^{t-k:t+l}|G^{0:t-k}, \boldsymbol{c}^{0:t-k}) = P(\boldsymbol{Y}^{t-k:t-k+l}|G^{t-k}, \boldsymbol{c}^{t-k})$ where $l$ is the length of the prediction.*

Then, we can have the following lemma:

**Lemma 4.1.** *With Assumptions 4.1 and 4.2, we have:*

$$
\begin{aligned}
\mathrm{P}\left(\boldsymbol{Y}^t \mid G^{0:t-1}\right) = \int \mathrm{P}\left(\boldsymbol{Y}^t \mid \boldsymbol{c}^{t-1}, G^{t-1}\right) \cdot \\
\mathrm{P}\left(\boldsymbol{c}^{t-1} \mid \boldsymbol{c}^{t-k}, G^{t-k:t-1}\right) \cdot \mathrm{P}\left(\boldsymbol{c}^{t-k} \mid G^{0:t-k}\right) d\boldsymbol{c}^{t-1} d\boldsymbol{c}^{t-k}.
\end{aligned}
\tag{3}
$$

The proof of Lemma 4.2 can be found in Appendix. From Lemma 4.1, we decompose the probability $\mathrm{P}\left(\boldsymbol{Y}^t \mid G^{0:t-1}\right)$ into three terms. Specifically, the last term necessitates encoding context information based on the historical trajectory $G^{0:t-k}$. The second term aims to update the context vector according to the recent trajectory $G^{t-k:t-1}$. The first term suggests using the context variable in conjunction with the current states to make predictions for the next timestamp. Besides making a single next-time prediction, our model can also predict trajectories $(\boldsymbol{Y}^{t-k}, \boldsymbol{Y}^{t-k+1}, \cdots, \boldsymbol{Y}^t)$ by modifying Eqn. 14.

Consequently, we divide each training sequence into two parts, namely $[0, t-k]$ and $(t-k, t]$ as in [19, 18]. The first part is used to encode contexts and nodes in the system, while the second part serves for updating and prediction purposes.

## 4.2 Context Acquirement from Spatio-temporal Signals

In this part, our goal is to acquire knowledge from the historical trajectory, i.e., $\{G^0, \cdots, G^{t-k}\}$ to encode contexts and nodes for initialization. To be specific, we first construct a temporal graph to capture spatial and temporal signals simultaneously. Subsequently, we employ the attention mechanism to update temporal node representations, which will be used to initialize the context representation and node representations for $\{G^{t-k}, \cdots, G^T\}$.

To begin, we construct a temporal graph containing two types of edges, i.e., spatial and temporal edges. Spatial edges are built when the distance between two nodes at the same timestamp is less than a threshold while temporal edges are between every two consecutive observations for each node. Specifically, in the constructed temporal graph $G^H$, there are $N(t-k+1)$ nodes $\{i_s\}^{s\in[0:t-k], i\in V}$

in total. The adjacent matrix $\boldsymbol{A}$ contains both spatial and temporal edges as follows:

$$\boldsymbol{A}(i^s, j^{s'}) = \begin{cases} exp(-d^s(i,j)) & s = s', exp(-d^s(i,j)) < \tau, \\ 1 & i = j, s' = s+1, \\ 0 & \text{otherwise}, \end{cases} \quad (4)$$

where $d^s(i,j)$ denotes the distance between particles $i$ and $j$ at timestamp $s$ and $\tau$ is the predefined threshold. Then, we utilize an attention-based GNN to extract spatio-temporal relationships into node representations. Here, we first compute the interaction scores between each node in $G^H$ with its neighboring nodes, and then aggregate their embeddings at the previous layer. Let $d$ represent the hidden dimension, the interaction score between $i^s$ and $j^{s'}$ at layer $l$ is:

$$w^{(l)}(i^s, j^{s'}) = \boldsymbol{A}(i^s, j^{s'})(\boldsymbol{W}_{query}\boldsymbol{h}_i^{s,(l)}) \star (\boldsymbol{W}_{key}\boldsymbol{h}_j^{s',(l)}), \quad (5)$$

where $\boldsymbol{W}_{query} \in \mathbb{R}^{d \times d}$ and $\boldsymbol{W}_{key} \in \mathbb{R}^{d \times d}$ are two matrices to map temporal node representations into different spaces. $\star$ computes the cosine similarity between two vectors. With the interaction scores, we can compute temporal node representations at the next layer:

$$\boldsymbol{h}_i^{s,(l+1)} = \boldsymbol{h}_i^{s,(l)} + \sigma\left(\sum_{j^{s'} \in \mathcal{N}(i^s)} w^{(l)}(i^s, j^{s'})\boldsymbol{W}_{value}\boldsymbol{h}_j^{s',(l)}\right), \quad (6)$$

where $\boldsymbol{W}_{value}$ is a weight matrix, $\mathcal{N}(i^s)$ collects all the neighbours of $i^s$ and $\sigma(\cdot)$ is an activation function. After stacking $L$ layers, we add temporal encoding and then summarize all these temporal node representations to initialize node representations for the upcoming ODE module:

$$\boldsymbol{q}_i^s = \boldsymbol{h}_i^{s,(L)} + \text{TE}(s), \quad \boldsymbol{u}_i^{t-k} = \frac{1}{t-k+1}\sum_{s=0}^{t-k} \sigma(\boldsymbol{W}_{sum}\boldsymbol{q}_i^s), \quad (7)$$

where $\text{TE}(s)[2i] = \sin\left(\frac{s}{10000^{2i/d}}\right)$, $\text{TE}(s)[2i+1] = \cos\left(\frac{s}{10000^{2i/d}}\right)$ and $\boldsymbol{W}_{sum}$ denotes a projection matrix. The initial context variable $\boldsymbol{c}^{t-k}$ is driven by summarizing all node representations:

$$\beta_i^t = tanh((\frac{1}{|V|}\sum_{i' \in V}\boldsymbol{u}_{i'}^{t-k})\boldsymbol{W}_{context}) \cdot \boldsymbol{u}_i^{t-k}, \quad \boldsymbol{c}^{t-k} = \sum_{i \in V}\beta_i^t\boldsymbol{u}_i^{t-k}, \quad (8)$$

where $\boldsymbol{W}_{context}$ is a learnable matrix and $\beta_i^t$ calculates the attention score for each node.

### 4.3 Context-attended Graph ODE

In this module, to model continuous evolution, we incorporate an ODE system into our approach. The precondition requires assuming that both the context variable and node representations are continuous to fit neural ODE models, which inherently holds for common dynamical systems in practice. We then introduce coupled ODEs to model the dynamic evolution of node representations and the context variable. Specifically, the context variable can be inferred during the evolution of node representations, which in turn drives the evolution of the system. We first introduce the assumption:

**Assumption 4.3.** *(Continuous) We assume that both context variable $\boldsymbol{c}^s$ and node representations $\boldsymbol{v}_i^s$ are continuously differentiable with respect to $s$.*

Then, to utilize the context variable and the current state for making future predictions, we introduce a graph ODE model. Let $\hat{\boldsymbol{A}}^s$ denote the adjacency matrix at timestamp $s$ with self-loop, we have:

$$\frac{d\boldsymbol{v}_i^s}{ds} = \Phi([\boldsymbol{v}_1^s, \cdots, \boldsymbol{v}_N^s, \boldsymbol{c}^s]) = \sigma(\sum_{j \in \mathcal{N}^s(i)} \frac{\hat{\boldsymbol{A}}_{ij}^s}{\sqrt{\hat{D}_i^s \cdot \hat{D}_j^s}}\boldsymbol{v}_j^s\boldsymbol{W}_1 + \boldsymbol{c}^s\boldsymbol{W}_2), \quad (9)$$

where $\mathcal{N}^s(i)$ denotes the neighbours of node $i$ at timestamp $s$ and $\hat{D}_i^s$ represents the degree of node $i$ according to $\hat{\boldsymbol{A}}^s$. The first term in Eqn. 9 aggregates information from its instant neighbors and the second term captures information from the current context information.

The next question is how to model the evolution of $\boldsymbol{c}^s$. Notice that we have:

$$\text{P}\left(\boldsymbol{c}^t \mid \boldsymbol{c}^{t-k}, G^{t-k:t}\right) = \int P(\boldsymbol{c}^t|\boldsymbol{c}^{t-\Delta t}, G^{t-\Delta t:t})\cdots$$
$$P(\boldsymbol{c}^{t-k+\Delta t}|\boldsymbol{c}^{t-k}, G^{t-k:t-k+\Delta t})d\boldsymbol{c}^{t-k+\Delta t}\cdots d\boldsymbol{c}^{t-\Delta t}, \quad (10)$$

where $\Delta t$ denotes a small time interval. With Assumption 4.3, we can simplify $P(\boldsymbol{c}^{t-k+\Delta t}|\boldsymbol{c}^{t-k}, G^{t-k:t-k+\Delta t})$ into $P(\boldsymbol{c}^{t-k+\Delta t}|\boldsymbol{c}^{t-k}, \boldsymbol{V}^{t-k}, d\boldsymbol{V}^{t-k})$ where $\boldsymbol{V}^{t-k}$ denotes the node embedding matrix at timestamp $t-k$ and $d\boldsymbol{V}^{t-k}$ is its differentiation. On this basis, we introduce another ODE to update the context variable as:

$$\frac{d\boldsymbol{c}^s}{ds} = \Phi^c(\text{AGG}(\{\boldsymbol{v}_i^s\}_{i \in V}), \text{AGG}(\{\frac{d\boldsymbol{v}_i^s}{ds}\}_{i \in V}), \boldsymbol{c}^s]), \tag{11}$$

where $\Phi^c$ is an MLP with the concatenated input and $\text{AGG}(\cdot)$ is an operator to summarize node representations such as averaging and sum. Compared to previous methods, the key to our CARE is to take into account the mutual impact between the environment and the trajectories, and model their evolution simultaneously by coupling Eqn. 9 and Eqn. 11. We also provide a theoretical analysis of the uniqueness of the solution to our system. To simplify the analysis, we set $\text{AGG}(\cdot)$ to summation and rewrite Eqn. 11 with learnable matrices $\boldsymbol{W}_3$, $\boldsymbol{W}_4$ and $\boldsymbol{W}_5$ as:

$$\frac{d\boldsymbol{c}^s}{ds} = \sigma\left(\sum_{i=1}^{N}(\boldsymbol{v}_i^s \boldsymbol{W}_3 + \frac{d\boldsymbol{v}_i^s}{ds}\boldsymbol{W}_4) + \boldsymbol{c}^s \boldsymbol{W}_5\right). \tag{12}$$

Then, we introduce the following assumption:

**Assumption 4.4.** *All time-dependent coefficients in Eqn. 9, i.e $\boldsymbol{A}_{ij}^t$, $\hat{D}_i^t$ are continuous with respect to $t$ and bounded by a constant $C > 0$. All parameters in the weight matrix are also bounded by a constant $W > 0$.*

With Assumption 4.4, we can deduce the following lemma:

**Lemma 4.2.** *Given the initial state $(t_0, \boldsymbol{v}_1^{t_0}, \cdots, \boldsymbol{v}_N^{t_0}, \boldsymbol{c}^{t_0})$, we claim that there exists $\varepsilon > 0$, s.t. the ODE system 9 and 12 has a unique solution in the interval $[t_0 - \varepsilon, t_0 + \varepsilon]$.*

The proof of Lemma 4.2 can be found in Appendix. Our theoretical analysis indicates that at least locally, the future system trajectory and context information are predictable based on their historical values [51], which is also an important property for dynamical system modeling [5, 28].

### 4.4 Decoder and Optimization

**Decoder.** We introduce an MLP $\Phi^d(\cdot)$ to predict both the position and velocity vectors using corresponding node representations, i.e., $[\hat{\boldsymbol{p}}_i^s, \hat{\boldsymbol{q}}_i^s] = \Phi^d(\boldsymbol{v}_i^s)$.

**Dynamic Graph Updating.** We can estimate the instant distance between nodes using the encoder and then construct the graphs, which could suffer from a large computational burden. To improve the efficiency of graph construction during ODE propagation, we not only update the graph structure every $\Delta s$, and but also introduce a graph updating strategy that calculates the distance between first-order and second-order neighbors in the last graph. By doing so, we can delete edges between first-order neighbors and add edges between second-order neighbors, reducing quadratic complexity to linear complexity in sparse graphs. We will also validate this empirically.

**Learning Objective.** Given the ground truth, we first minimize the mean squared error (MSE) of the predicted trajectory. Moreover, we require both node and context representations to be robust to noise attacks to improve the robustness of the ODE system. The overall objective is written as:

$$\mathcal{L} = \sum_{s=t-k}^{t} ||\hat{\boldsymbol{Y}}^s - \boldsymbol{Y}^s|| + \eta(||\tilde{\boldsymbol{V}}^s - \boldsymbol{V}^s|| + ||\tilde{\boldsymbol{c}}^s - \boldsymbol{c}^s||), \tag{13}$$

where $\hat{\boldsymbol{Y}}^s$ denotes the predictions from the encoder and $\eta$ is a parameter set to $0.1$ to balance two losses. $\tilde{\boldsymbol{V}}^s$ and $\tilde{\boldsymbol{c}}^s$ denote the perturbed representations under noise attack to the input.

## 5 Experiments

We evaluate our proposed CARE on both particle-based and mesh-based physical systems. To ensure the accuracy of our results, we use a rigorous data split strategy, where first $80\%$ of the samples are reserved for training purposes and the remaining $10\%$ are set aside for testing and validating,

Table 1: The RMSE ($\times 10^{-2}$) results of the compared methods with the prediction lengths 1, 5, 10 and 20. $v_x$, $v_y$ and $v_z$ represent the velocity in the direction of each coordinate axis.

| Prediction Length | +1 | | | +5 | | | +10 | | | +20 | | |
|---|---|---|---|---|---|---|---|---|---|---|---|---|
| Variable | $v_x$ | $v_y$ | $v_z$ | $v_x$ | $v_y$ | $v_z$ | $v_x$ | $v_y$ | $v_z$ | $v_x$ | $v_y$ | $v_z$ |
| *Lennard-Jones Potential* | | | | | | | | | | | | |
| LSTM | 3.95 | 3.92 | 3.68 | 9.12 | 9.21 | 9.15 | 10.84 | 10.87 | 10.76 | 14.82 | 14.94 | 14.67 |
| GNS | 3.28 | 3.75 | 3.39 | 7.97 | 8.05 | 7.68 | 10.09 | 10.15 | 10.13 | 13.65 | 13.62 | 13.59 |
| STGCN | 2.91 | 3.08 | 2.95 | 5.06 | 5.17 | 5.11 | 6.89 | 6.90 | 6.93 | 9.31 | 9.32 | 9.44 |
| MeshGraphNet | 2.89 | 3.13 | 2.94 | 5.29 | 5.53 | 5.28 | 7.03 | 7.09 | 7.11 | 9.12 | 9.21 | 9.24 |
| CG-ODE | 1.79 | 2.05 | 1.71 | 3.47 | 3.92 | 3.38 | 5.46 | 5.99 | 5.36 | 9.03 | 9.26 | 8.92 |
| TIE | 1.62 | 1.98 | 1.47 | 3.25 | 3.90 | 3.15 | 5.24 | 5.82 | 5.17 | 8.24 | 8.34 | 8.47 |
| Ours | **0.76** | **0.89** | **1.01** | **2.94** | **3.16** | **2.85** | **5.01** | **4.69** | **4.71** | **5.75** | **5.91** | **5.82** |
| *3-body Stillinger-Weber Potential* | | | | | | | | | | | | |
| LSTM | 17.11 | 17.14 | 17.18 | 23.64 | 23.69 | 23.60 | 25.46 | 25.42 | 25.48 | 28.44 | 28.45 | 28.44 |
| GNS | 15.39 | 15.27 | 15.33 | 22.14 | 22.19 | 22.17 | 25.29 | 25.36 | 25.31 | 27.18 | 27.15 | 27.14 |
| STGCN | 12.33 | 12.31 | 12.35 | 17.94 | 17.96 | 17.91 | 20.08 | 20.14 | 20.13 | 23.49 | 23.51 | 23.52 |
| MeshGraphNet | 12.16 | 12.10 | 12.13 | 18.33 | 18.38 | 18.34 | 20.65 | 20.62 | 20.71 | 23.62 | 23.54 | 23.61 |
| CG-ODE | 9.78 | 9.74 | 9.75 | 12.11 | 12.05 | 12.14 | 15.55 | 15.58 | 15.50 | 16.17 | 16.24 | 16.22 |
| TIE | 10.18 | 10.26 | 10.19 | 14.75 | 14.70 | 14.73 | 18.42 | 18.45 | 18.41 | 20.92 | 21.04 | 21.36 |
| Ours | **4.21** | **4.29** | **4.18** | **9.74** | **9.79** | **9.71** | **13.65** | **13.71** | **13.57** | **15.30** | **15.39** | **15.35** |

CARE

Ground Truth

Figure 2: Visualization of *Lennard-Jones Potential* with multiple timestamps. We render the 3D positions of each particle according to the historical positions and predicted velocities.

separately. During training, we split each trajectory sample into two parts, i.e., a conditional part and a prediction part. We initialize node representations and the context representation based on the first part and utilize the second part to supervise the model. The size of the two parts is represented as conditional length and prediction length, respectively. Our approach is compared with various baselines for interacting systems modeling, i.e., LSTM [15], STGCN [67], GNS [45], MeshGraphNet [42], TIE [46] and CG-ODE [19].

## 5.1 Performance on Particle-based Physical Simulations

**Datasets.** We evaluate our proposed CARE on two particle-based simulation datasets with temporal environmental variations, i.e., *Lennard-Jones Potential* [47] and *3-body Stillinger-Weber Potential* [4]. *Lennard-Jones Potential* is popular in modeling electronically neutral atoms or molecules. *3-body Stillinger-Weber Potential* provides more complex relationships in atom systems The temperature in two particle-based systems is continuously changed along with the time to model the environmental variations. The objective is to predict the future velocity values in all directions, i.e., $v_x$, $v_y$ and $v_z$. More details can be found in Appendix.

**Performance Comparison.** We evaluate the performance in terms of RMSE with different prediction lengths. Table 1 show the compared results on these two datasets. We can observe that our proposed CARE outperforms all the baselines on two datasets. In particular, compared with TIE, CARE accomplishes an error reduction of 24.03% and 36.35% on two datasets, respectively. The remarkable performance can be attributed to two factors: (1) Introduction of the context variable. Our CARE infers the context states during the evolution of the system, which can help the model understand environmental variations. (2) Introduction of robust learning. We add noise attack to both system and context states, which improves the model generalization to potential distribution changes.

Table 2: The RMSE results of the compared methods over different prediction lengths 1, 10, 20 and 50. $v_x$, $v_y$ and $p$ represent the velocity in different directions and the pressure field, respectively.

| Prediction Length | +1 | | | +10 | | | +20 | | | +50 | | |
|---|---|---|---|---|---|---|---|---|---|---|---|---|
| Variable | $v_x$ | $v_y$ | $p$ | $v_x$ | $v_y$ | $p$ | $v_x$ | $v_y$ | $p$ | $v_x$ | $v_y$ | $p$ |
| *CylinderFlow* | | | | | | | | | | | | |
| LSTM | 3.35 | 29.4 | 12.5 | 7.06 | 44.8 | 17.8 | 9.47 | 49.5 | 19.9 | 14.3 | 73.6 | 42.3 |
| GNS | 3.12 | 28.8 | 11.9 | 7.18 | 44.3 | 17.3 | 9.01 | 49.6 | 19.2 | 13.5 | 73.2 | 41.6 |
| STGCN | 2.68 | 26.7 | 11.0 | 5.47 | 42.1 | 16.9 | 6.72 | 45.6 | 18.4 | 9.15 | 68.7 | 40.0 |
| MeshGraphNet | 1.75 | 22.4 | 10.6 | 4.09 | 39.7 | 15.7 | 5.38 | 44.5 | 17.2 | 7.92 | 64.3 | 37.7 |
| CG-ODE | 1.05 | 20.4 | 8.51 | 3.44 | 36.8 | 13.6 | 4.15 | 38.5 | 17.1 | 5.14 | 61.2 | 32.3 |
| TIE | 1.22 | 20.8 | 8.94 | 3.75 | 35.2 | 13.0 | 4.62 | 40.6 | 16.0 | 5.87 | 59.5 | 32.1 |
| Ours | **0.87** | **19.1** | **7.21** | **3.02** | **32.9** | **11.8** | **3.95** | **37.8** | **13.9** | **4.97** | **55.8** | **29.4** |
| *Airfoil* | | | | | | | | | | | | |
| LSTM | 7.49 | 7.73 | 1.92 | 8.86 | 9.02 | 3.78 | 10.8 | 11.0 | 4.71 | 14.9 | 15.7 | 4.96 |
| GNS | 6.95 | 7.14 | 1.69 | 8.20 | 8.34 | 3.34 | 10.2 | 10.5 | 3.98 | 14.2 | 14.1 | 4.11 |
| STGCN | 6.24 | 5.35 | 1.07 | 6.57 | 6.51 | 2.33 | 7.88 | 8.01 | 3.16 | 11.6 | 11.8 | 3.17 |
| MeshGraphNet | 4.72 | 4.68 | 0.50 | 5.89 | 5.74 | 1.23 | 6.32 | 6.48 | 1.85 | 9.03 | 9.12 | 2.08 |
| CG-ODE | 4.26 | 4.32 | 0.35 | 4.78 | 4.70 | 0.46 | 5.81 | 5.66 | 1.04 | 7.39 | 7.85 | 1.69 |
| TIE | 4.17 | 4.39 | 0.33 | 4.99 | 4.86 | 0.51 | 5.75 | 5.62 | 0.95 | 7.25 | 7.63 | 1.44 |
| Ours | **3.51** | **4.11** | **0.19** | **3.86** | **3.75** | **0.34** | **4.16** | **4.12** | **0.45** | **6.74** | **6.82** | **0.81** |

Figure 3: Visualization of the CylinderFlow Dataset with multiple timestamps. We render the velocity in the $x$-axis in the fluid field of our CARE and the ground truth.

**Visualization.** Figure 2 visualizes the prediction of positions in comparison to the ground truth on *Lennard-Jones Potential*. Here, we sample six timestamps in every trajectory to validate the performance of both short-term and long-term predictions. From the qualitative results, we can observe that in the first three frames, the particle motion is not strenuous due to low temperature in the system. Surprisingly, our proposed CARE can always make faithful physical simulations close to the ground truth even though the system environment is highly variable.

## 5.2 Performance on Mesh-based Physical Simulations

**Datasets.** We employ two popular mesh-based simulation datasets, i.e., *CylinderFlow*, and *Airfoil*. *CylinderFlow* consists of simulation data from modeling an incompressible flow governed by the Navier-Stokes equations. Notably, the initial flow velocity of the incoming water flow to the cylinder varies cyclically over time, meaning the Reynolds number of the flow field also changes periodically. *Airfoil* is generated in a similar manner through simulations of a compressible flow, wherein the inlet velocity over the wing varies cyclically over time. We aim to forecast the velocity values $v_x$ and $v_y$, as well as the pressure $p$. More details can be found in Appendix.

**Performance Comparison.** The performance with respect to different variables is recorded in Table 2. From the results, we can observe that the average performance of the proposed CARE is over the best baseline by 12.99% and 22.78% on two datasets, respectively. Note that unsteady flow [11, 16] is a crucial problem in recent fluid dynamics, our proposed CARE can benefit abundant complex mesh-based simulations under environmental variations.

**Visualization.** Moreover, we show the qualitative results of the best baseline and our CARE in comparison to the ground truth. From the results, we can observe that our CARE can capture more accurate signals in unsteady fluid dynamics. In particular, in the last two frames with complicated

Table 3: Ablation study on four datasets.

| Datasets | Lennard-Jones | | | 3-body Stillinger-Weber | | | CylinderFlow | | | Airfoil | | |
|---|---|---|---|---|---|---|---|---|---|---|---|---|
| Variable | $v_x$ | $v_y$ | $v_z$ | $v_x$ | $v_y$ | $v_z$ | $v_x$ | $v_y$ | $p$ | $v_x$ | $v_y$ | $p$ |
| CARE V1 | 6.98 | 7.12 | 7.06 | 18.2 | 18.3 | 18.3 | 6.13 | 60.4 | 32.2 | 7.13 | 7.21 | 1.43 |
| CARE V2 | 6.03 | 6.35 | 6.30 | 16.8 | 16.5 | 16.6 | 5.21 | 57.2 | 29.8 | 6.94 | 6.99 | 1.15 |
| **Ours** | **5.75** | **5.91** | **5.82** | **15.3** | **15.4** | **15.4** | **4.97** | **55.8** | **29.4** | **6.74** | **6.82** | **0.81** |

Figure 4: (a), (b) The performance with respect to different condition and prediction lengths on CylinderFlow and Airfoil. (c) The sensitivity of interval on Lennard-Jones Potential (LJP) and 3-body Stillinger-Weber Potential (SWP) datasets. (d) The comparison of running time for our dynamic graph updating and full pairwise calculation on two particle-based datasets.

structures, our CARE still can generate superior simulations in both scenarios under potential environmental variation while the baseline fails, which shows the superiority of our proposed CARE.

## 5.3 Further Analysis

**Ablation Study.** To analyze the effectiveness of different components in our CARE, we introduce two different variants: (1) CARE V1, which removes the context variable in Eqn. 9; (2) CARE V2, which removes the robust learning term in Eqn. 13. The compared performance is recorded in Figure 3 and we have two observations. First, our full model outperforms CARE V1, which indicates the incorporation of the context variable would benefit interacting system modeling under temporal environmental variation. Second, without the robust learning term, the performance would get worse, implying that improving the robustness can also benefit tackling the distribution changes.

**Parameter Sensitivity.** We begin by analyzing the performance with respect to different condition lengths and prediction lengths. Here the condition length and prediction length vary from {10,15,20,25,30}, {20,50}, respectively. From the results in Figure 4 (a) and (b), we can observe that our proposed CARE can always achieve superior performance compared with CG-ODE. Moreover, we can observe that a longer condition length would benefit the performance in most cases due to more provided information. It can also be seen that a smaller interval for graph updating would improve the performance before saturation from Figure 4 (c).

**Efficiency.** To show the efficiency of our proposed dynamic graph updating, we propose a model variant named CARE E, which calculates all pairwise distances to update graph structure during the evolution. The computational cost is recorded in Figure 4 (d) and we can observe that our strategy can reduce a large number of computational costs, which validates the complexity analysis before.

## 6 Conclusion

This paper studies the problem of modeling interacting dynamics under temporal environmental variation and we propose a probabilistic framework to depict the dynamical system. Then, a novel approach named CARE is proposed. CARE first constructs an encoder to initialize the context variable indicating the environment and then utilizes a coupled ODE system, which combines both the context variable and node representation to drive the evolution of the system. Finally, we introduce both efficient dynamical graph updating and robust learning strategies to enhance our framework. Extensive experiments on four datasets validate the superiority of our CARE.

**Broader Impacts and Limitations.** This work presents an effective learning-based model for interacting dynamical systems under temporal environmental variation, which can benefit complex physical simulations such as unsteady flow. Moreover, our study provides a new perspective on

modeling environmental variations for fluid dynamics and intermolecular interactions. One potential limitation is that our CARE cannot directly fit more physical scenarios requiring abundant external knowledge. In future work, we would extend our CARE to more complicated applications such as rigid dynamics.

## Acknowledgement

This work was partially supported by NSF 2211557, NSF 1937599, NSF 2119643, NSF 2303037, NSF 2312501, NASA, SRC Jump 2.0, Okawa Foundation Grant, Amazon Research Awards, Cisco research grant, Picsart Gifts, and Snapchat Gifts.

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

# A  Proof of Lemma 4.1

**Lemma 4.1.** *With Assumptions 4.1 and 4.2, we have:*

$$
\begin{aligned}
\mathrm{P}\left(\boldsymbol{Y}^t \mid G^{0:t-1}\right) = \int \mathrm{P}\left(\boldsymbol{Y}^t \mid \boldsymbol{c}^{t-1}, G^{t-1}\right) \cdot \\
\mathrm{P}\left(\boldsymbol{c}^{t-1} \mid \boldsymbol{c}^{t-k}, G^{t-k:t-1}\right) \cdot \mathrm{P}\left(\boldsymbol{c}^{t-k} \mid G^{0:t-k}\right) d\boldsymbol{c}^{t-1} d\boldsymbol{c}^{t-k}.
\end{aligned}
\tag{14}
$$

*Proof.* We have:

$$
\begin{aligned}
&\mathrm{P}\left(\boldsymbol{Y}^t \mid G^{0:t-1}\right) \\
&= \int \mathrm{P}\left(\boldsymbol{Y}^t \mid \boldsymbol{c}^{t-1}, G^{0:t-1}\right) \cdot \mathrm{P}\left(\boldsymbol{c}^{t-1} \mid G^{0:t-1}\right) d\boldsymbol{c}^{t-1} \\
&= \int \mathrm{P}\left(\boldsymbol{Y}^t \mid \boldsymbol{c}^{t-1}, G^{t-1}\right) \cdot \mathrm{P}\left(\boldsymbol{c}^{t-1} \mid G^{0:t-1}\right) d\boldsymbol{c}^{t-1} \\
&= \int \mathrm{P}\left(\boldsymbol{Y}^t \mid \boldsymbol{c}^{t-1}, G^{t-1}\right) \cdot \mathrm{P}\left(\boldsymbol{c}^{t-1} \mid \boldsymbol{c}^{t-k}, G^{0:t-1}\right) \cdot \mathrm{P}\left(\boldsymbol{c}^{t-k} \mid G^{0:t-k}\right) d\boldsymbol{c}^{t-1} d\boldsymbol{c}^{t-k} \\
&= \int \mathrm{P}\left(\boldsymbol{Y}^t \mid \boldsymbol{c}^{t-1}, G^{t-1}\right) \cdot \mathrm{P}\left(\boldsymbol{c}^{t-1} \mid \boldsymbol{c}^{t-k}, G^{t-k:t-1}\right) \cdot \mathrm{P}\left(\boldsymbol{c}^{t-k} \mid G^{0:t-k}\right) d\boldsymbol{c}^{t-1} d\boldsymbol{c}^{t-k}
\end{aligned}
\tag{15}
$$

$\square$

# B  Proof of Lemma 4.2

For convenience, Eqn. 9 in the main paper is repeated as:

$$
\frac{d\boldsymbol{v}_i^s}{ds} = \Phi([\boldsymbol{v}_1^s, \cdots, \boldsymbol{v}_N^s, \boldsymbol{c}^s]) = \sigma\Big( \sum_{j \in \mathcal{N}^s(i)} \frac{\hat{A}_{ij}^s}{\sqrt{\hat{D}_i^s \cdot \hat{D}_j^s}} \boldsymbol{v}_j^s \boldsymbol{W}_1 + \boldsymbol{c}^s \boldsymbol{W}_2 \Big),
\tag{16}
$$

Eqn. 12 is repeated as:

$$
\frac{d\boldsymbol{c}^s}{ds} = \sigma\left( \sum_{i=1}^N (\boldsymbol{v}_i^s \boldsymbol{W}_3 + \frac{d\boldsymbol{v}_i^s}{ds} \boldsymbol{W}_4) + \boldsymbol{c}^s \boldsymbol{W}_5 \right).
\tag{17}
$$

**Lemma 4.2.** *Given the initial state $(t_0, \boldsymbol{v}_1^{t_0}, \cdots, \boldsymbol{v}_N^{t_0}, \boldsymbol{c}^{t_0})$, we claim that there exists $\varepsilon > 0$, s.t. the ODE system Eqn. 16 and Eqn. 17 has a unique solution in the interval $[t_0 - \varepsilon, t_0 + \varepsilon]$.*

To begin, we introduce the Picard–Lindelöf Theorem as follows:

**Theorem B.1.** *(Picard–Lindelöf Theorem) Let $D \subseteq \mathbb{R} \times \mathbb{R}^n$ be a closed rectangle with $(t_0, y_0) \in D$. Let $f : D \to \mathbb{R}^n$ be a function which is continuous with respect to $t$ and Lipschitz continuous with respect to $y$. Then, there exists some $\varepsilon > 0$ such that the initial value problem:*

$$
y'(t) = f(t, y(t)), \quad y(t_0) = y_0.
\tag{18}
$$

*has a unique solution $y(t)$ in the interval $[t_0 - \varepsilon, t_0 + \varepsilon]$.*

*Proof.* Let $\boldsymbol{A}_{ij}^s = 0$ if $j \notin \mathcal{N}^s(i)$ and denote $\boldsymbol{M}_{ij}^s = \frac{\hat{A}_{ij}^s}{\sqrt{\hat{D}_i^s \cdot \hat{D}_j^s}}$. Then, we transpose them with Eqn. 16 and 17 becoming:

$$
\begin{aligned}
\frac{d(\boldsymbol{v}_i^s)^T}{ds} &= \sigma\left( \sum_{j=1}^N \boldsymbol{M}_{ij}^s \boldsymbol{W}_1^T (\boldsymbol{v}_j^s)^T + \boldsymbol{W}_2^T (\boldsymbol{c}^s)^T \right), \\
\frac{d(\boldsymbol{c}^s)^T}{ds} &= \sigma\left( \sum_{i=1}^N (\boldsymbol{W}_3^T (\boldsymbol{v}_i^s)^T + \boldsymbol{W}_4^T \frac{d(\boldsymbol{v}_i^s)^T}{ds}) + \boldsymbol{W}_5^T (\boldsymbol{c}^s)^T \right).
\end{aligned}
\tag{19}
$$

Let $\boldsymbol{Y}^s = \begin{pmatrix} (\boldsymbol{v}_1^s)^T \\ \vdots \\ (\boldsymbol{v}_N^s)^T \\ (\boldsymbol{c}^s)^T \end{pmatrix} \in \mathbf{R}^{(N \times d_v + d_c) \times 1}$, where $\boldsymbol{v}_i \in \mathbf{R}^{d_v}, \boldsymbol{c} \in \mathbf{R}^{d_c}$.

From the Eqn. 19 we get the ODE system $\frac{d\boldsymbol{Y}^s}{ds} = f(\boldsymbol{Y}^s; \theta)$ with fixed parameters $\theta$ for the ODE solver. Here the function $f(\boldsymbol{Y}^t; \theta)$ is continuous w.r.t $t$ since all components in the vector $\boldsymbol{Y}^t$ are continuous w.r.t $t$ and $\theta$ does not depend on t.

Now consider activation functions $\sigma$, such as ReLU, that satisfy the following inequality for all $x$ and $y$:
$$\|\sigma(x) - \sigma(y)\| \le \|x - y\|$$
Then, for any two solutions $\boldsymbol{Y}_1^s, \boldsymbol{Y}_2^s$, we have:

$$\|f(\boldsymbol{Y}_1^s; \theta) - f(\boldsymbol{Y}_2^s; \theta)\|_2 \le \left\| \begin{pmatrix} \delta_1 \\ \vdots \\ \delta_N \\ \delta_{N+1} \end{pmatrix} \right\|_2,$$

$$\delta_i = \sum_{j=1}^N \boldsymbol{M}_{ij}^s \boldsymbol{W}_1^T \Big[ (\boldsymbol{v}_{1j}^s)^T - (\boldsymbol{v}_{2j}^s)^T \Big] + \boldsymbol{W}_2^T \Big[ (\boldsymbol{c}_1^s)^T - (\boldsymbol{c}_2^s)^T \Big], \quad i = 1, \dots, N$$

$$\delta_{N+1} = \sum_{i=1}^N \left( \boldsymbol{W}_3^T \Big[ (\boldsymbol{v}_{1i}^s)^T - (\boldsymbol{v}_{2i}^s)^T \Big] + \boldsymbol{W}_4^T \Big[ \sigma(\frac{d(\boldsymbol{v}_{1i}^s)^T}{ds}) - \sigma(\frac{d(\boldsymbol{v}_{2i}^s)^T}{ds}) \Big] \right) + \boldsymbol{W}_5^T \Big[ (\boldsymbol{c}_1^s)^T - (\boldsymbol{c}_2^s)^T \Big].$$

To simplify the representation, denote:
$$\Delta \boldsymbol{v}_j^s = \boldsymbol{v}_{1j}^s - \boldsymbol{v}_{2j}^s, \quad \Delta \boldsymbol{c}^s = \boldsymbol{c}_1^s - \boldsymbol{c}_2^s.$$

Then by triangular inequality, we have $\forall i \in \{1, \cdots, N\}$,

$$\big\|\delta_i^T\big\|_2^2 = \|\sum_{j=1}^N \Delta \boldsymbol{v}_j^s \boldsymbol{W}_1 (\boldsymbol{M}_{ij}^s)^T + \Delta \boldsymbol{c}^s \boldsymbol{W}_2\|_2^2$$

$$\le (\sum_{j=1}^N \|\Delta \boldsymbol{v}_j^s \boldsymbol{W}_1 (\boldsymbol{M}_{ij}^s)^T\|_2 + \|\Delta \boldsymbol{c}^s \boldsymbol{W}_2\|_2)^2$$

$$\le (WM \sum_{j=1}^N \|\Delta \boldsymbol{v}_j^s\|_2 + W\|\Delta \boldsymbol{c}^s\|_2)^2,$$

$$\big\|\delta_{N+1}^T\big\|_2^2 = \|\sum_{i=1}^N \left( \Delta \boldsymbol{v}_i^s \boldsymbol{W}_3 + [\sigma(\frac{d\boldsymbol{v}_{1i}^s}{ds}) - \sigma(\frac{d\boldsymbol{v}_{2i}^s}{ds})] \boldsymbol{W}_4 \right) + \Delta \boldsymbol{c}^s \boldsymbol{W}_5\|_2^2$$

$$\le (\sum_{i=1}^N \|\Delta \boldsymbol{v}_i^s \boldsymbol{W}_3\|_2 + \sum_{i=1}^N \|[\sigma(\frac{d\boldsymbol{v}_{1i}^s}{ds}) - \sigma(\frac{d\boldsymbol{v}_{2i}^s}{ds})] \boldsymbol{W}_4\|_2 + \|\Delta \boldsymbol{c}^s \boldsymbol{W}_5\|_2)^2$$

$$\le (W \sum_{i=1}^N \|\Delta \boldsymbol{v}_i^s\|_2 + W \sum_{i=1}^N \|\frac{d\boldsymbol{v}_{1i}^s}{ds} - \frac{d\boldsymbol{v}_{2i}^s}{ds}\|_2 + W\|\Delta \boldsymbol{c}^s\|_2)^2$$

$$= (W \sum_{i=1}^N \|\Delta \boldsymbol{v}_i^s\|_2 + W \sum_{i=1}^N \|\delta_i^T\|_2 + W\|\Delta \boldsymbol{c}^s\|_2)^2$$

$$\le \left(W \sum_{i=1}^N \|\Delta \boldsymbol{v}_i^s\|_2 + WN(WM \sum_{j=1}^N \|\Delta \boldsymbol{v}_j^s\|_2 + W\|\Delta \boldsymbol{c}^s\|_2) + W\|\Delta \boldsymbol{c}^s\|_2\right)^2$$

$$= \left((W + MNW^2) \sum_{i=1}^N \|\Delta \boldsymbol{v}_i^s\|_2 + (W + NW^2)\|\Delta \boldsymbol{c}^s\|_2\right)^2.$$

Without loss of generality, we assume constants $M, W > 1$. Thus, we have the following result:

$$
\begin{aligned}
\|f(\boldsymbol{Y}_1^s; \theta) - f(\boldsymbol{Y}_2^s; \theta)\|_2^2 &\leq \left\|(\delta_1^T, \cdots, \delta_N^T, \delta_{N+1}^T)\right\|_2^2, \\
&= \left\|\delta_1^T\right\|_2^2 + \cdots + \left\|\delta_N^T\right\|_2^2 + \left\|\delta_{N+1}^T\right\|_2^2 \\
&\leq N(WM\sum_{j=1}^{N}\|\Delta\boldsymbol{v}_j^s\|_2 + W\|\Delta\boldsymbol{c}^s\|_2)^2 + \\
&\quad \left((W + MNW^2)\sum_{i=1}^{N}\|\Delta\boldsymbol{v}_i^s\|_2 + (W + NW^2)\|\Delta\boldsymbol{c}^s\|_2\right)^2 \\
&\leq \left[NW^2M^2 + (W + MNW^2)^2\right]\left(\sum_{j=1}^{N}\|\Delta\boldsymbol{v}_j^s\|_2 + \|\Delta\boldsymbol{c}^s\|_2\right)^2 \\
&= \left[NW^2M^2 + (W + MNW^2)^2\right]\|\boldsymbol{Y}_1^s - \boldsymbol{Y}_2^s\|_2^2.
\end{aligned}
$$

Therefore the function $f$ is L-Lipschitz with $L = \sqrt{NW^2M^2 + (W + MNW^2)^2}$. By Picard–Lindelöf theorem, we prove the uniqueness of the solution. $\square$

## C   Dataset Details

We evaluate our proposed CARE on four physical simulation datasets with temporal environmental variation. All these four datasets involve at least one thousand nodes. Then we introduce the details of these four datasets.

- *Lennard-Jones Potential* (a.k.a. 6-12 potential) is popular in modeling electronically neutral atoms or molecules, which can be formulated as:

$$
V_{\text{LJ}} = 4\varepsilon\left[\left(\frac{\sigma}{r}\right)^{12} - \left(\frac{\sigma}{r}\right)^6\right], \tag{20}
$$

where $r$ is the distance between particle pairs, $\sigma$ denotes the size of the particle, $\epsilon$ denotes the depth of the potential well. The first term denotes the attractive force, which decreases as the distance between particles increases. The second term denotes the repulsive force, which increases when two particles are too close. The temperature in the system is changed along with the time to model the environmental variations and a high temperature would bring a more intense molecular motion.

- *3-body Stillinger-Weber Potential* provides more complex relationships besides pairwise relationships in *Lennard-Jones Potential*. It contains both two-body and three-body terms with the following formulation:

$$
V_{\text{SW}} = \sum_i\sum_{j>i}\phi_2\left(r_{ij}\right) + \sum_i\sum_{j\neq i}\sum_{k>j}\phi_3\left(r_{ij}, r_{ik}, \theta_{ijk}\right), \tag{21}
$$

where $\phi_2\left(r_{ij}\right) = A_{ij}\epsilon_{ij}\left[B_{ij}\left(\frac{\sigma_{ij}}{r_{ij}}\right)^{p_{ij}} - \left(\frac{\sigma_{ij}}{r_{ij}}\right)^{q_{ij}}\right]\exp\left(\frac{\sigma_{ij}}{r_{ij} - a_{ij}\sigma_{ij}}\right)$ is the two-body term and $\phi_3\left(r_{ij}, r_{ik}, \theta_{ijk}\right) = \lambda_{ijk}\epsilon_{ijk}\left[\cos\theta_{ijk} - \cos\theta_{0ijk}\right]^2\exp\left(\frac{\gamma_{ij}\sigma_{ij}}{r_{ij} - a_{ij}\sigma_{ij}}\right)\exp\left(\frac{\gamma_{ik}\sigma_{ik}}{r_{ik} - a_{ik}\sigma_{ik}}\right)$ is the three-body term. The two body term is similar to *Lennard-Jones Potential* to model the pairwise relationships and the three body term can consider the angles among atom triplets. Similarly, the temperature is changed along with the time to model the environmental variations and a high temperature would also bring a more intense molecular motion.

- *CylinderFlow* is a popular computational fluid dynamics (CFD) simulation dataset, which models the fluid flow around a given cylinder by OpenFoam [22]. It consists of simulation data from modeling an incompressible flow governed by the Navier-Stokes equations. The Reynolds number, denoted by Re, is a dimensionless quantity that characterizes the flow regime of the fluid. It is defined as: $Re = \frac{\rho V D}{\mu}$, where $\rho$ is the density of the fluid, $V$ is the velocity of the fluid relative to the cylinder, $D$ is the diameter of the cylinder, and $\mu$ is the dynamic viscosity of the fluid. The transition from the laminar to turbulent flow usually happens at a critical Reynolds number, which

depends on the geometry of the cylinder and the properties of the fluid. In general, the flow is more likely to be laminar for small cylinders and viscous fluids, and more likely to be turbulent for large cylinders and low-viscosity fluids. Notably, the initial flow velocity $V$ of the incoming water flow to the cylinder varies cyclically over time, meaning the Reynolds number of the flow field also changes periodically.

- *Airfoil* is generated in a similar manner through simulations of a compressible flow by Open-Foam [22]. The lift coefficient of an airfoil relies on a number of factors, e.g., the angle of attack, the shape of the airfoil, and the Reynolds number of the flow. The angle of attack is the one between the chord line of the airfoil (the straight line connecting the leading and trailing edges) and the direction of the incoming flow. The Reynolds number, as mentioned in the previous question, is a dimensionless quantity that characterizes the flow regime of the fluid. It also plays a crucial role in the lift produced by an airfoil, as it determines whether the flow around the airfoil is laminar or turbulent. For laminar flow, the air moves smoothly over the surface of the airfoil, while for turbulent flow, it moves in a chaotic, swirling pattern. The Reynolds number is given by: $Re = \frac{\rho V c}{\mu}$, where $c$ is the chord length of the airfoil, and $\mu$ denotes the viscosity of the fluid. The lift coefficient is typically higher for laminar flow than for turbulent flow, up to a certain point where the flow separates from the airfoil. Notably, in our simulation datasets, the inlet velocity $V$ over the wing also varies cyclically over time.

## D  Details of Baselines

Our proposed method is compared with a range of competing baselines as follows:

- LSTM [15] is a widely recognized approach for sequence prediction problems. It involves three gates, i.e., forget gate, input gate and output gate, enabling the model to acquire knowledge of long-term relationships.
- STGCN [67] is a deep learning approach to handle spatial dependencies and temporal dynamics in complicated spatio-temporal data. It involves a recurrent component and a message passing component for effective analysis of spatio-temporal signals.
- GNS [45] utilizes a graph to represent a physical dynamical system and then utilizes a message passing neural network to explore complicated dynamics and interactions among multiple objects.
- MeshGraphNet [42] characterize each physical system as meshes, followed by graph neural networks to learn interacting dynamics. Moreover, remeshing techniques are adopted to fit the multi-resolution nature in irregular meshes.
- CG-ODE [19] models both nodes and edges jointly through two groups of ODEs, which can capture the evolution of both objective and interaction in the system.
- TIE [46] attempts to improve the particle-based simulations by decomposing edges into both ends, and introducing abstract nodes to capture global information in the system.

## E  Algorithm

The whole learning algorithm of CARE is summarized in Algorithm 1.

## F  Implementation Details

To ensure the accuracy of our results, we use a rigorous data split strategy, where first $80\%$ of the samples are reserved for training purposes and the remaining $10\%$ are set aside for testing and validating, separately. Following [19], we also ensure that no sequence overlap exists on training, validation and testing sets. In particular, each of these particle-based datasets consists of $14400$ training trajectories, $1800$ validation trajectories, and $1800$ test trajectories while $7200$ training samples, $900$ validation samples and $900$ test trajectories are for mesh-based datasets. During training, we split each trajectory sample into two parts, i.e., a conditional part and a prediction part. We initialize node representations and the context representation based on the first part and utilize the second part to supervise the model. The size of the two parts is represented as conditional length and prediction length, respectively. We would vary two lengths to show the performance comprehensively.

**Algorithm 1** Learning Algorithm of the proposed CARE

---

**Input:** The training sequences $\left\{G^0, G^1, \cdots, G^t\right\}$.
**Output**: The parameters in our CARE.

---

 1: Initialize the parameters in our model;
 2: **while** not convegence **do**
 3:    **for** each training sequence **do**
 4:       Divide the sequence into two segments;
 5:       Build the temporal graph using Eqn. 4;
 6:       Initialize both node representations and context representations for ODEs using Eqn. 8;
 7:       Solve the coupled ODEs, i.e., Eqns. 9 and 11;
 8:       Add noise into the input for perturbed hidden states;
 9:       Feed these hidden states into a decoder $\Phi^d(\cdot)$ to get the predictions;
10:       Calculate the loss in Eqn. 13;
11:       Update the parameters in CARE using back propagation;
12:    **end for**
13: **end while**

---

In our implementation, we utilize the sum operator as AGG in Eqn. 11. To solve the ODE systems on a time grid which is five times denser than the observed time steps, we employ the fourth-order Runge-Kutta method as in the torchdiffeq Python package [24], using PyTorch [40]. We also use the adjoint method [5] to reduce memory usage. All experiments are conducted on a single NVIDIA A100 GPU. We set the latent dimension to 256 and the dropout rate to 0.2. For optimization, we use the Adam optimizer with weight decay by mini-batch stochastic gradient descent, setting the learning rate to 0.01. Overall, our proposed CARE offers a novel approach for modeling and predicting complex systems with multiple interacting objects.

# G  More Experiment Results

## G.1  Model Comparison

We first compare our CARE with MP-NODE [10], which is an ODE-based approach for homogeneous dynamical systems. The compared result on particle-based simulation datasets and mesh-based simulation datasets are recorded in Table 4 and Table 5, respectively. From the results, we can validate the superiority of our CARE in tackling the temporal environmental variation and making accurate long-term predictions.

Table 4: Results on particle-based physical simulations with the prediction lengths 1, 5, 10 and 20. $v_x$, $v_y$ and $v_z$ represent the velocity in the direction of each coordinate axis.

| Prediction Length | +1 | | | +5 | | | +10 | | | +20 | | |
|---|---|---|---|---|---|---|---|---|---|---|---|---|
| Variable | $v_x$ | $v_y$ | $v_z$ | $v_x$ | $v_y$ | $v_z$ | $v_x$ | $v_y$ | $v_z$ | $v_x$ | $v_y$ | $v_z$ |
| *Lennard-Jones Potential* | | | | | | | | | | | | |
| MP-NODE | 1.45 | 1.79 | 1.41 | 3.08 | 3.74 | 3.02 | 5.36 | 5.91 | 5.26 | 8.46 | 8.36 | 8.95 |
| Ours | **0.76** | **0.89** | **1.01** | **2.94** | **3.16** | **2.85** | **5.01** | **4.69** | **4.71** | **5.75** | **5.91** | **5.82** |
| *3-body Stillinger-Weber Potential* | | | | | | | | | | | | |
| MP-NODE | 9.95 | 9.82 | 9.87 | 12.47 | 12.41 | 12.49 | 16.05 | 16.14 | 16.10 | 17.06 | 17.15 | 17.09 |
| Ours | **4.21** | **4.29** | **4.18** | **9.74** | **9.79** | **9.71** | **13.65** | **13.71** | **13.57** | **15.30** | **15.39** | **15.35** |

## G.2  Results with Different Prediction Lengths

To evaluate the performance of the proposed CARE in different settings, we vary the prediction length and compare the performance of different approaches. In particular, we show the results with the prediction length $\{8, 15\}$ on two particle-based simulation datasets in Table 6. The results with the prediction length $\{15, 30\}$ on two mesh-based simulation datasets are shown in Table 7. From the compared results, we can validate the superiority of our CARE in various settings.

Table 5: Results on mesh-based physical simulations over different prediction lengths 1, 10, 20 and 50. $v_x$, $v_y$ and $p$ represent the velocity in different directions and the pressure field, respectively.

| Prediction Length | +1 | | | +10 | | | +20 | | | +50 | | |
|---|---|---|---|---|---|---|---|---|---|---|---|---|
| Variable | $v_x$ | $v_y$ | $p$ | $v_x$ | $v_y$ | $p$ | $v_x$ | $v_y$ | $p$ | $v_x$ | $v_y$ | $p$ |
| *CylinderFlow* | | | | | | | | | | | | |
| MP-NODE | 1.11 | 20.6 | 8.62 | 3.68 | 37.2 | 13.8 | 4.36 | 38.8 | 17.7 | 5.59 | 61.8 | 32.7 |
| Ours | **0.87** | **19.1** | **7.21** | **3.02** | **32.9** | **11.8** | **3.95** | **37.8** | **13.9** | **4.97** | **55.8** | **29.4** |
| *Airfoil* | | | | | | | | | | | | |
| MP-NODE | 4.41 | 4.44 | 0.38 | 4.85 | 4.76 | 0.49 | 5.89 | 5.72 | 1.23 | 7.45 | 7.97 | 1.78 |
| Ours | **3.51** | **4.11** | **0.19** | **3.86** | **3.75** | **0.34** | **4.16** | **4.12** | **0.45** | **6.74** | **6.82** | **0.81** |

Table 6: Results on particle-based physical simulations with the prediction lengths 8 and 15. $v_x$, $v_y$ and $v_z$ represent the velocity in the direction of each coordinate axis.

| Prediction Length | +8 | | | +15 | | |
|---|---|---|---|---|---|---|
| Variable | $v_x$ | $v_y$ | $v_z$ | $v_x$ | $v_y$ | $v_z$ |
| *Lennard-Jones Potential* | | | | | | |
| LSTM | 9.44 | 9.40 | 9.57 | 12.68 | 12.75 | 12.61 |
| GNS | 8.86 | 8.92 | 8.85 | 11.99 | 11.84 | 12.08 |
| STGCN | 6.28 | 6.24 | 6.33 | 7.85 | 8.01 | 8.09 |
| MeshGraphNet | 6.47 | 6.42 | 6.55 | 7.99 | 8.08 | 8.14 |
| CG-ODE | 5.33 | 5.16 | 5.17 | 7.94 | 7.59 | 7.71 |
| TIE | 4.97 | 4.68 | 4.71 | 7.65 | 7.28 | 7.44 |
| Ours | **4.85** | **4.17** | **4.55** | **5.52** | **5.19** | **5.06** |
| *3-body Stillinger-Weber Potential* | | | | | | |
| LSTM | 22.89 | 22.93 | 22.90 | 26.74 | 26.79 | 26.78 |
| GNS | 22.36 | 22.38 | 22.31 | 26.04 | 26.05 | 26.02 |
| STGCN | 17.79 | 17.88 | 17.83 | 21.48 | 21.42 | 21.46 |
| MeshGraphNet | 17.92 | 17.84 | 17.95 | 21.86 | 21.82 | 21.84 |
| CG-ODE | 13.61 | 13.68 | 13.63 | 17.11 | 17.15 | 17.08 |
| TIE | 16.79 | 16.76 | 16.81 | 20.04 | 20.08 | 20.02 |
| Ours | **11.94** | **11.97** | **11.88** | **14.89** | **14.95** | **14.81** |

### G.3 Visualization

Moreover, we show more visualization of our proposed CARE and the best baseline (i.e., TIE) compared with the ground truth. In particular, given that we have shown the time steps in $\{1, 100, 200, 300, 400, 500\}$ in Figure 3 and Figure 4, now we select them in $\{5, 150, 250, 350, 450\}$ for Lennard-Jones Potential and CylinderFlow, respectively. The compared results are shown in Figure 5 and Figure 6. We can observe that our CARE can generate more accurate trajectories compared with the baselines in most cases, which validates the superiority of our CARE again. For example, in the last row in Figure 5, our CARE can accurately recover the velocity distribution around the cylinder while the baseline fails.

## H  More Related Work

**Graph Neural Networks.** Graph Neural Networks (GNNs) have garnered significant success for their remarkable capabilities in graph representation learning [26, 62, 72, 33, 13], which is integral to a variety of downstream applications, including node classification [26], link prediction [71], and graph classification [62]. These approaches typically employ the message-passing mechanism, enabling the iterative updating of node representations with the aid of neighboring information. Recently, GNNs have successfully branched out into modeling interactive dynamics [42, 46, 45]. For

Table 7: Results on mesh-based physical simulations over different prediction lengths 15 and 30. $v_x$, $v_y$ and $p$ represent the velocity in different directions and the pressure field, respectively.

| Prediction Length | +15 | | | +30 | | |
|---|---|---|---|---|---|---|
| Variable | $v_x$ | $v_y$ | $p$ | $v_x$ | $v_y$ | $p$ |
| *CylinderFlow* | | | | | | |
| LSTM | 8.25 | 47.43 | 17.62 | 11.09 | 59.86 | 33.24 |
| GNS | 8.16 | 47.95 | 17.54 | 11.36 | 60.49 | 33.72 |
| STGCN | 6.27 | 44.37 | 16.97 | 7.03 | 56.18 | 31.45 |
| MeshGraphNet | 5.13 | 42.05 | 14.65 | 6.54 | 52.96 | 28.63 |
| CG-ODE | 3.92 | 37.45 | 13.79 | 5.28 | 47.12 | 23.69 |
| TIE | 4.07 | 37.91 | 13.72 | 5.31 | 47.17 | 23.65 |
| Ours | **3.48** | **35.6** | **12.9** | **4.26** | **44.9** | **20.7** |
| *Airfoil* | | | | | | |
| LSTM | 8.75 | 8.84 | 3.96 | 12.86 | 18.74 | 4.78 |
| GNS | 8.14 | 8.02 | 3.51 | 11.52 | 11.44 | 4.04 |
| STGCN | 7.64 | 7.38 | 2.79 | 10.14 | 10.16 | 3.14 |
| MeshGraphNet | 6.05 | 6.19 | 1.27 | 7.11 | 7.09 | 1.43 |
| CG-ODE | 5.68 | 5.53 | 0.81 | 6.82 | 6.88 | 1.21 |
| TIE | 5.41 | 5.33 | 0.79 | 6.64 | 6.71 | 1.19 |
| Ours | **4.08** | **4.02** | **0.41** | **4.95** | **5.11** | **0.62** |

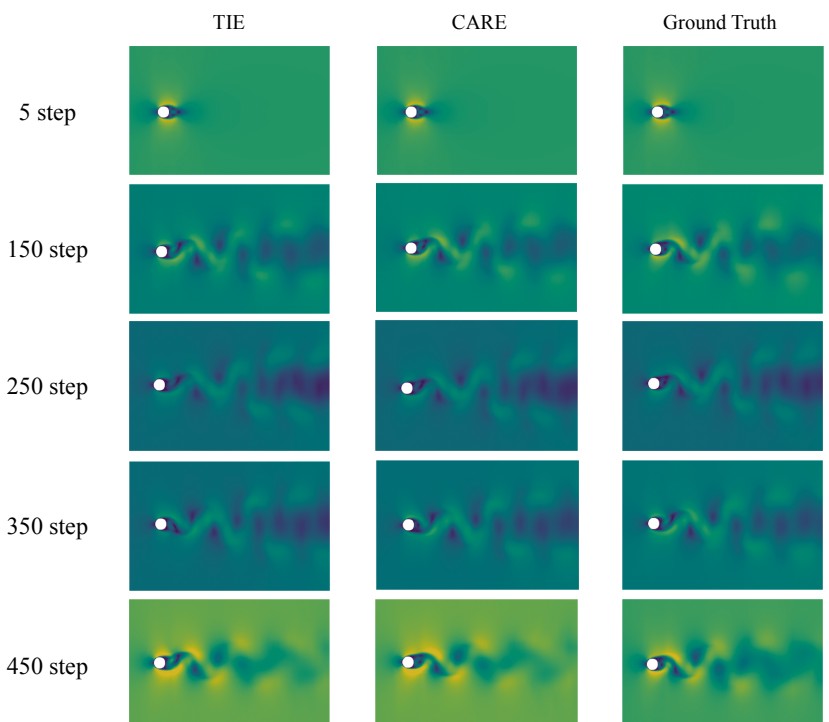

Figure 5: More visualization of velocity in CylinderFlow dataset with varying time steps in {5, 150, 250, 350, 450}.

instance, MeshGraphNet [42] employs a message passing neural network to facilitate the modeling of interactions between objectives, thereby outputting the next-time predictions. However, an inherent drawback lies in the inability of discrete GNNs to encapsulate the continuous nature of system

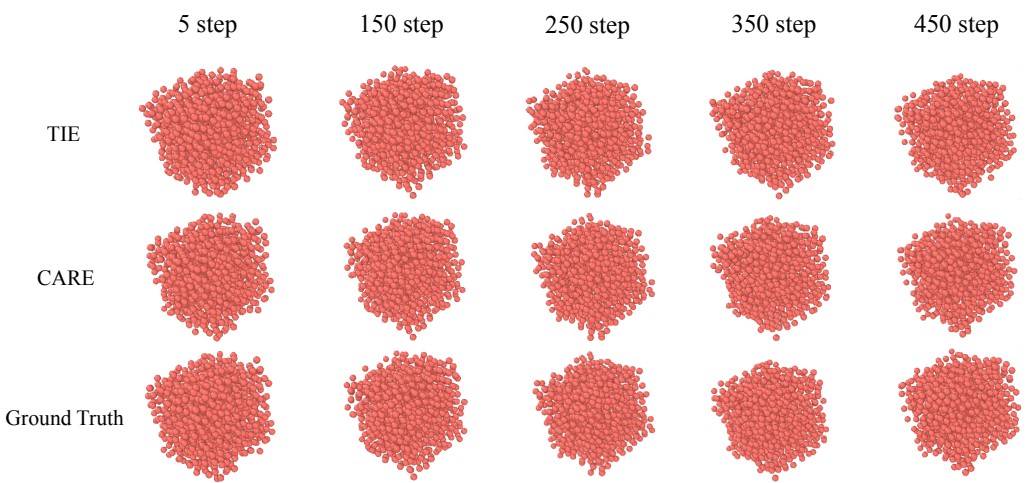

Figure 6: More visualization of Lennard-Jones Potential with varying time steps in {5, 150, 250, 350, 450}.

dynamics. To relieve this limitation, we present a novel graph-based ODE system named CARE for the modeling of interacting dynamics, which enriches the capabilities of making long-term predictions under potential environmental variation.

**Graph-based ODE.** Neural ODEs have been integrated into GNNs, resulting in the development of Graph ODEs that are applicable to both static and dynamic graphs. Graph ODEs on static graphs [60, 70] primarily aim to mitigate overfitting by formalizing derivatives using both initial and immediate node representations. Meanwhile, Graph ODEs on dynamic graphs are utilized for tasks such as traffic flow forecasting [9] and social analysis [19], demonstrating effective performance on irregularly sampled partial observation data. For example, STGODE [9] employs tensor computation to conduct continuous message passing, which facilitates accurate long-term predictions by overcoming the network depth limitations. Despite these advancements, existing works fall short of addressing the temporal environmental variation in interacting dynamics. To fill this gap, we propose a novel approach CARE to handle this problem by injecting a context variable in the Graph ODE system.

# I  Potential Negative Impacts

To the best of our knowledge, we have not found any negative impact of our work.

