# OpenReview forum: "CARE: Modeling Interacting Dynamics Under Temporal Environmental Variation"
_NeurIPS.cc/2023/Conference — NeurIPS 2023 poster_

### Official Review · Reviewer_QmzE · 2023-07-05

**Soundness:** 3 good
**Presentation:** 4 excellent
**Contribution:** 3 good
**Rating:** 6
**Confidence:** 3

**Summary:**

The paper aims to model out-of-distribution dynamics of environments with interacting entities using a seq-to-seq style graph neural net architecture. The proposed model attempts to capture the invariant aspects of the dynamical environment via a context embedding that follows a neural ordinary differential equation. The paper reports results on a number of physical systems modeling tasks, as well as some theoretical aspects of the proposed approach.

**Strengths:**

The paper is very well written. Figure 2 describes the methodology simply and clearly. The paper reports a large set of experiments and shows consistent improvement over a large set of benchmarks, though I think it misses the most essential ones (see my comments below). The paper exercises a systematic scientific writing approach where the key assumptions are pointed out and their most important analytical conclusions, such as model consistency, are analyzed, although with rather straightforward proof techniques.

**Weaknesses:**

The proposed method is novel per se. It is also intuitive and well-justified, but it appears to put together a number of existing tools in the most straightforward way one can think of. While I am convinced by the quality of the proposed solution, I am a bit skeptical about its scientific value, i.e. how exactly it enhances our knowledge base.

The conceptual novelty of the proposed method over some existing graph-based probabilistic ODE approaches such as IGP-ODE [60] and [Ref1] is not clarified. It is also not obvious to the reader why these methods that have suitable specs for handling out-of-distribution data, due to rigorous uncertainty modeling, should not be among the list of models in comparison. For instance IGP-ODE also model interaction dynamics, it reports disentanglement results with the motivation of out-of-distribution detection, and it also takes the dynamics of context variables to its center.

[Ref1] Look et al., Cheap and Deterministic Inference for Deep State-Space Models of Interacting Dynamical Systems.

**Questions:**

Although the paper motivates the proposed methodology with the primary application of generalization over out-of-distribution cases, I am not able to point out how the proposed experiments test this aspect and according to which metric. The results tables appear to quantify model performance only in in-distribution use cases, i.e. both training and test data are assumed to follow the same dynamics. Is my understanding correct? If not, how exactly is the out-of-distribution data acquired and what is the actual difference between the train and test splits that makes the test observations out-of-distribution?

Since the answers to the above questions play a critical role in my evaluation, for now I set my grade to borderline. It is likely to swing based on further interactions with the authors and other reviewers.


**Limitations:**

Section 6 of the paper discusses the limitations of the proposed approach in sufficient detail.

---

> ### Author Rebuttal · Authors · 2023-08-10
>
> We are truly grateful for the time you have taken to review our paper and your insightful review. Here we address your comments in the following.
>
> > Q1. The proposed method is novel per se. It is also intuitive and well-justified, but it appears to put together a number of existing tools in the most straightforward way one can think of. While I am convinced by the quality of the proposed solution, I am a bit skeptical about its scientific value, i.e. how exactly it enhances our knowledge base.
>
> A1. Thanks for your acknowledgement of our novelty. Our method is totally based on the proposed probabilistic model, which could make our method a little straightforward. The scientific value of our proposed method is two-fold:
> - **Real-world Applications.** Our approach is tailored to dynamical systems affected by fluctuating environments. Such scenarios are prevalent in real-world applications. Examples include particle-based systems experiencing variable temperatures or unsteady flows influenced by different Reynolds numbers. By incorporating context variances, we can tackle these complex and real-world dynamical systems.
>
> - **Disentangling Representations.** A salient feature of our method is its ability to distinguish between object-specific and context representations. This disentanglement offers deeper insights into the impact of shifting environments on dynamical systems. We believe this perspective can serve as a crucial foundation for subsequent research in the area.
>
>
>
> > Q2. The conceptual novelty of the proposed method over some existing graph-based probabilistic ODE approaches such as IGP-ODE [60] and [Ref1] is not clarified. It is also not obvious to the reader why these methods that have suitable specs for handling out-of-distribution data, due to rigorous uncertainty modeling, should not be among the list of models in comparison. For instance IGP-ODE also model interaction dynamics, it reports disentanglement results with the motivation of out-of-distribution detection, and it also takes the dynamics of context variables to its center.
>
> A2. Thanks for your comment. We first compare these two methods with the proposed methods. IGP-ODE combines latent Gaussian process with ODEs while GDSSM [ref1] utilizes the graph neural networks with a Gaussian mixture model. The differences between these two works and ours lie in three points:
> - **Different Objectives**. Our method primarily hones in on addressing the intricacies of dynamical systems in the presence of fluctuating environments. In contrast, both IGP-ODE and GDSSM, do not encompass this specific challenge.
> - **Different Methodology**: Our methods introduce a context variable to model the dynamics of environments based on a probabilistic model while these two methods focus on modeling the dynamics of objects using Gaussian processes and Gaussian Mixture Model, respectively.
> - **Different Efficiency**. Our method is tailored to efficiently cater to both particle-based and mesh-based systems, even when these encompass a vast array of nodes. Conversely, the models like IGP-ODE and GDSSM concentrate on vehicular and kinematic systems, leveraging intricate modeling techniques through Gaussian processes and the Gaussian Mixture Model.
>
>
> In our practical experiments, both these methods faced Out-Of-Memory challenges in our setup. Specifically, these models encountered OOM issues with datasets exceeding 30 nodes, whereas our datasets consistently feature thousands of nodes.
>
>
> | Number of nodes | IGP-ODE   | GDSSM    |
> |-----------------|-----------|----------|
> | 10              | 23.57GB   | 19.65GB  |
> | 20              | 62.54GB   | 68.23GB  |
> | 30              | OOM       | OOM      |
>
>
>
> > Q3. Although the paper motivates the proposed methodology with the primary application of generalization over out-of-distribution cases, I am not able to point out how the proposed experiments test this aspect and according to which metric. The results tables appear to quantify model performance only in in-distribution use cases, i.e. both training and test data are assumed to follow the same dynamics. Is my understanding correct? If not, how exactly is the out-of-distribution data acquired and what is the actual difference between the train and test splits that makes the test observations out-of-distribution?
>
> A3. Thanks for your comment. Indeed, in particle-based systems, individual samples often exhibit different initial temperatures and laws of change, leading to distinct dynamics. As a result, the training and test data inherently possess varying dynamic contexts, which in turn introduces different dynamics. Furthermore, within individual samples, the ever-changing temperature introduces temporal distribution shifts, thereby causing the data distribution to evolve over time. It is this temporal variability and context-driven change that we refer to when discussing "out-of-distribution." To ensure greater clarity, we will rephrase our approach as addressing "context-varying dynamics" to better convey the challenges which our method seeks to tackle.
>
> In light of these responses, we hope we have addressed your concerns, and hope you will consider raising your score. If there are any additional notable points of concern that we have not yet addressed, please do not hesitate to share them, and we will promptly attend to those points.
>
> **Reference**
>
> [Ref1] Look et al., Cheap and Deterministic Inference for Deep State-Space Models of Interacting Dynamical Systems.

---

> > ### Comment · Reviewer_QmzE · 2023-08-12
> > **Concerns addressed**
> >
> > Thanks, my concerns have been addressed, the major one being the conceptual confusion about OOD. Also interesting additional result about the memory footprint of the alternative methods. I raise my score to 6 to acknowledge the contribution, not higher to account for the slight incremental nature of it.

---

> > > ### Author Response · Authors · 2023-08-12
> > > **Thanks for your feedback and raising the score!**
> > >
> > > Thanks again for your feedback and increasing the rating! We are pleased to know that our responses have addressed your concerns. We really appreciate your efforts on reviewing our paper, your insightful comments and support.

---

### Official Review · Reviewer_P3pg · 2023-07-05

**Soundness:** 3 good
**Presentation:** 3 good
**Contribution:** 3 good
**Rating:** 7
**Confidence:** 4

**Summary:**

The paper proposes a new model architecture for modeling multi-agent dynamics. The model consists of three parts (encoder, dynamics, decoder) where the encoder initializes the latent state of each agent and the context, the dynamics is modeled via NeuralODEs and the decoder is a standard MLP. New to the paper is that the model explicitly consider a context variable that evolves over time.

**Strengths:**

### New Architecture
The explicit modeling of the context variable and its temporal evolution is new to me. I like the idea and the experiments support its benefits.

### Experiments
The experiments in the paper are exhaustive and show the advantage of the new method compared to competitors. I also like the ablation study that shows the merits of the individual contributions.


**Weaknesses:**

I think the paper slightly oversells what it is doing:

### Temporal distribution shift
The paper is called "modeling interacting dynamics under temporal distribution shift" but there is very little about temporal distribution shift in the data. I would have expected that for instance the temperature in Sec. 5.1 is different between training and test data. If this is the case, the authors should state it more explicitly in the experiments. If not, I would suggest to tone it down a bit.

### Probabilistic Modeling
 I like the new model architecture and it seems principled to me. However, the paper states in the abstract "provide a probabilistic view for out-of-distribution dynamics", whlie the learnt model is completely deterministic. I would also suggest here to not overstate the contribution.


**Questions:**

### Interpretability
Can you please comment on the interpretability of the context variable? I think it is not and this should also be clearly stated in the manuscript or demonstrated otherwise.

### Naive solutions
I could also come up with a naive solution in which the context variable is modeled as N+1-th object in the graph that is connected to all other variables but does not have any observations. How would that compare to the aforementioned solution?
How does a static context variable compare to the dynamic solution? Static context variables have for instance also been considered in [1] which should be cited.

### References
[1] Yıldız, Çağatay, Melih Kandemir, and Barbara Rakitsch. "Learning interacting dynamical systems with latent Gaussian process ODEs." Advances in Neural Information Processing Systems 35 (2022): 9188-9200.



**Limitations:**

None.

---

> ### Author Rebuttal · Authors · 2023-08-10
>
> We are truly grateful for the time you have taken to review our paper, your insightful comments and support. Your positive feedback is incredibly encouraging for us! In the following response, we would like to address your major concern and provide additional clarification.
>
> > Q1: The paper is called "modeling interacting dynamics under temporal distribution shift" but there is very little about temporal distribution shift in the data. I would have expected that for instance the temperature in Sec. 5.1 is different between training and test data. If this is the case, the authors should state it more explicitly in the experiments. If not, I would suggest to tone it down a bit.
>
> A1. Thanks for your comment. Indeed, in particle-based systems, individual samples often exhibit different initial temperatures and laws of change, leading to distinct dynamics. As a result, the training and test data inherently possess varying dynamic contexts, which in turn introduces different dynamics. We acknowledge the potential lack of clarity on this aspect in the paper and will also tone it down a bit to make it more clear.
>
> > Q2: I like the new model architecture and it seems principled to me. However, the paper states in the abstract "provide a probabilistic view for out-of-distribution dynamics", while the learnt model is completely deterministic. I would also suggest here to not overstate the contribution.
>
> A2. Thanks for your comment. We take your suggestion and will adjust the phrasing to more accurately state "provide a view for context-varying dynamics". We value your feedback and strive for clarity and accuracy in our work.
>
> > Q3: Can you please comment on the interpretability of the context variable? I think it is not and this should also be clearly stated in the manuscript or demonstrated otherwise.
>
> A3. Thanks for your comment. We have visualized the first 10 dimensions of the context variable in Figure A. From this visualization, we can find that the context variable experiences rapid changes over time. This variation supports our assertion that a dynamic context variable is crucial for adapting to the shifting environments found in dynamical systems. Moreover, the complex dynamics of the context variable indicates that the variance of environments (e.g., temperature) could bring in complicated impacts to the dynamical systems.
>
> > Q4. I could also come up with a naive solution in which the context variable is modeled as N+1-th object in the graph that is connected to all other variables but does not have any observations. How would that compare to the aforementioned solution? How does a static context variable compare to the dynamic solution? Static context variables have for instance also been considered in [1] which should be cited.
>
> A4. Thanks for your comment. Following your suggestion, we have added two model variants below:
> - CARE-O, which includes the N+1-th object in the graph connected to all other variables;
> - CARE-S, which utilizes a static context variable instead.
>
> The compared performance on two datasets is recorded as below. From the results, we can obtain that the full model perform better than CARE-O, which shows that directly combines heterogeneous nodes, i.e., objects and contexts on a simple graph could overlook critical information, e.g., the gradients of node representations in Eq. 11.
> Moreover, we can find that models relying on static context variables, i.e., CARE-S performs worse since it is hard to capture the dynamic nature of varied environments. We will definitely cite [1] in our revised version.
>
> ****
> | Dataset  | Lennard-Jones Potential |  	|  	| CylinderFlow |  	|  	|
> |----------|:-----------------------:|:----:|:----:|:------------:|:----:|:----:|
> | Variable |       	$v_x$       	|  $v_y$ |  $v_z$ |  	$v_x$ 	|  $v_y$ |   $p$  |
> | CARE-O   |       	6.69      	| 6.80 | 6.73 | 	4.27 	| 38.4 | 14.2 |
> | CARE-S  |       	7.96      	| 8.18 | 8.01 | 	5.32 	| 39.6 | 15.8 |
> | CARE 	|       	5.75      	| 5.91 | 5.82 | 	3.95 	| 37.8 | 13.9 |
>
> Thanks again for appreciating our work and for your constructive suggestions. Please let us know if you have further questions.

---

> > ### Comment · Reviewer_P3pg · 2023-08-12
> > **After rebuttal**
> >
> > Hi all,
> >
> > I thank the authors for adressing my questions, and, particularly, for running the additional experiments. I have no further questions.

---

> > > ### Author Response · Authors · 2023-08-12
> > > **Thanks again for your feedback!**
> > >
> > > Thanks again for your feedback! We are pleased to know that our responses have addressed your concerns. We really appreciate your efforts on reviewing our paper, your insightful comments and support.

---

### Official Review · Reviewer_ZaqZ · 2023-07-06

**Soundness:** 3 good
**Presentation:** 4 excellent
**Contribution:** 4 excellent
**Rating:** 7
**Confidence:** 4

**Summary:**

The main idea of this work is to propose a novel approach called Context-attended Graph ODE (CARE) for modeling interacting dynamical systems under temporal distribution shift. The paper formalizes the problem of temporal distribution shift in interacting dynamics modeling and proposes a probabilistic framework to understand the relationships between trajectories and contexts. Experimental results demonstrate the exceptional performance of the CARE model, outperforming state-of-the-art methods in accurately predicting long-term trajectories, even in the presence of environmental variations.

**Strengths:**

- Novel approach: The CARE model introduced in this work presents a novel methodology by incorporating continuous context variations and system states into a coupled ODE system for modeling interacting dynamical systems under temporal distribution shift.
- Superior performance: Extensive experimental results demonstrate that the CARE model outperforms state-of-the-art approaches in accurately predicting long-term trajectories, even in the presence of environmental variations.
- Clarity: The paper effectively summarizes its contributions, including problem formalization, novel methodology, and comprehensive experiments. This clarity allows readers to readily grasp the significance of the research.
- Extensive experiments: The paper conducts a thorough evaluation of the CARE model by performing experiments on diverse dynamical systems. The results highlight the superiority of the proposed approach compared to state-of-the-art methods. Additionally, the paper includes an ablation study and a parameter sensitivity analysis to further validate the effectiveness of the CARE model.

**Weaknesses:**

I don't see any weakness in the paper.

**Questions:**

- Where is Lemma 4.2?
- How to set the threshold mentioned in Line 159?
- Will the dynamic graph updating with larger $\delta s$ leads to many indirect connections and how will it affect the results?
- How is the performance of proposed CARE on dynamical systems with more nodes, let's say, 10, 100 and more?
- As shown in table 2, CARE still suffers from accumulated error on longer time-series. Do you have any idea on how to decrease the accumulated error?

**Limitations:**

The limitation is discussed.

---

> ### Author Rebuttal · Authors · 2023-08-10
>
> We are truly grateful for the time you have taken to review our paper, your insightful comments and support. Your positive feedback is incredibly encouraging for us! In the following response, we would like to address your major concern and provide additional clarification.
>
> > Q1: Where is Lemma 4.2?
>
> A1: Thanks for the comment. Lemma 4.2 can be found on line 206, and its corresponding proof is detailed in Appendix C.
>
> > Q2: How to set the threshold mentioned in Line 159?
>
> A2: Thanks for the comment. When constructing the graph structure, we set the threshold at the 30th percentile of all distances. For further validation of the robustness of this parameter choice, we experiment with varying percentiles in {25, 30, 35} on two datasets. The results are shown below, , which demonstrates the robustness of our parameter selection.
>
> | Percentile          	| 25 |  	|  	|  30 |  	|  	| 35|  	|  	|
> |-------------------------|:----:|:----:|:----:|:----:|:----:|:----:|:----:|------|------|
> | Lennard-Jones Potential |  	|  	|  	|  	|  	|  	|  	|  	|  	|
> | Variable            	| $v_x$  | $v_y$  | $v_z$  | $v_x$  | $v_y$  | $v_z$  | $v_x$  | $v_y$  | $v_z$  |
> | Ours                	| 5.98 | 6.07 | 6.05 | 5.75 | 5.91 | 5.82 | 5.93 | 6.06 | 6.02	|
> | CylinderFlow        	|  	|  	|  	|  	|  	|  	| 	 |  	|  	|
> | Variable            	|  $v_x$ |  $v_y$ |  $p$  |  $v_x$ |  $v_y$ |   $p$  |  $v_x$ |  $v_y$ |  $ p$  |
> | Ours                	| 4.13 | 38.2 | 13.9 | 3.95 | 37.8 | 13.9 | 4.08 | 37.9 | 13.9 |
>
>
> > Q3: Will the dynamic graph updating with larger $\Delta_s$ leads to many indirect connections and how will it affect the results?
>
> A3: Thanks for the comment. Indeed, an excessively large $\Delta_s$ might introduce a number of indirect connections, which in turn could potentially degrade performance. To empirically validate this, we have conducted experiments. As evidenced in Fig.5(c), it is clear that a large $\Delta_s$ (exceeding 20) would decrease the performance.
>
> > Q4: How is the performance of proposed CARE on dynamical systems with more nodes, let's say, 10, 100 and more?
>
> A4. Thanks for the comment. Actually, the experiments we conducted already involved systems with a significant number of nodes, well over the figures you have mentioned. Specifically, the datasets and their respective node numbers are as follows:
>
> | Dataset                        | Number of Nodes |
> |--------------------------------|-----------------|
> | Lennard-Jones Potential        | 1000            |
> | 3-body Stillinger-Weber Potential | 1000        |
> | CylinderFlow                   | 9200            |
> | Airfoil                        | 10720           |
>
> We'll ensure these details are more clearly stated in our revised manuscript.
>
> > Q5: As shown in table 2, CARE still suffers from accumulated error on longer time-series. Do you have any idea on how to decrease the accumulated error?
>
> A5. Thanks for the comment. Error accumulation is always a longstanding problem in long-term prediction. To mitigate this issue, we're considering several strategies for our future research:
> - **Enhanced Model Capacity**. We can develop more advanced graph ODE models with higher capacity, e.g., second-order ODE and augmented ODE. These advanced models could fit the spatial data better, which has the potential to decrease the accumulated error.
> - **Adaptive Settings**. We can utilize active learning and online learning to have more observations, which would provide valuable feedback to enable further fine-tuning and adjustments during long-term predictions.
> - **Ensemble Learning**. Incorporating multiple models with varying time horizons can be a promising strategy. By combining their strengths, we can potentially minimize both bias and variance of the prediction, further decreasing accumulated error.
>
> Thanks again for appreciating our work and for your constructive suggestions. Please let us know if you have further questions.

---

> > ### Comment · Reviewer_ZaqZ · 2023-08-11
> > **Thanks for the response**
> >
> > I would like to thank the authors for the response. My questions are fully addressed.

---

> > > ### Author Response · Authors · 2023-08-11
> > > **Thanks again for your feedback!**
> > >
> > > Thanks again for your feedback! We are pleased to know that our responses have addressed your concerns. We really appreciate your efforts on reviewing our paper, your insightful comments and support.

---

### Author Rebuttal · Authors · 2023-08-10

Dear Reviewers,

We thank you for your careful reviews and constructive suggestions. We acknowledge the positive comments such as "Novel approach" (Reviewer ZaqZ), “Superior performance” (Reviewer ZaqZ), "Clarity" (Reviewer ZaqZ), "Extensive experiments” (Reviewer ZaqZ), “I like the idea and the experiments support its benefits.” (Reviewer P3pg), “The experiments in the paper are exhaustive” (Reviewer P3pg), “The proposed method is novel per se.” (Reviewer QmzE), “The paper is very well written” (Reviewer QmzE), “reports a large set of experiments and shows consistent improvement” (Reviewer QmzE). We have also responded to your questions point by point.

The figure results are attached in the PDF. Please let us know if you have any follow-up questions. We will be happy to answer them.

Best regards,

the Authors

---

### Decision · Program_Chairs · 2023-09-21

**Decision:**

Accept (poster)

**Comment:**

The paper extends works on interacting dynamical systems to handle temporal distribution shift caused by environmental variations. The scenario present as well as the approach to handle it is novel, and the paper will be a useful contribution to the research community. The reviewers also agreed that the paper is well-written and proved its claims with extensive experiments. As such we decided that this paper should be accepted to this conference.